# Weather Preferences for Urban Tourism: An Empirical Study in the Greek Capital of Athens, Greece

**Dimitrios Kapetanakis** [1,2,*] , **Elena Georgopoulou** [1] , **Sevastianos Mirasgedis** [1] **and Yannis Sarafidis** [1]

1   Institute for Environmental Research and Sustainable Development, National Observatory of Athens,
I. Metaxas and Vassilis Pavlou, 15236 Penteli, Greece; elenag@noa.gr (E.G.); seba@noa.gr (S.M.);
sara@noa.gr (Y.S.)

2   FACE3TS S.A., 11471 Athens, Greece

*   Correspondence: dkapetan@noa.gr; Tel.: +30-69-7600-6071

**Abstract:** Several climate indices have been developed to analyze the relationship between climatic variables and tourist comfort at different destinations, although, none of the indices applied so far in cities have been informed by empirical data collected exclusively at urban tourist destinations. The present paper aims to cover this gap by developing an "Urban Climate Comfort Index" (UCCI) that integrates critical climate variables for urban tourism and is informed by empirical data from an in-situ survey conducted in southern Europe, namely, in close proximity to the Acropolis Museum in Athens, Greece. The survey provided input on the ideal and unacceptable climatic conditions as perceived by urban tourists and on the relevant weight of the selected climatic parameters. Tourist preferences were then translated into a numerical scale by assigning utility scores of 0% and 100% to the "unacceptable" and "ideal" values while using a linear change for the intermediate values. Hence, a best-fitting utility function for each climatic variable was created, and all utility functions were then aggregated through their relative weights to form the UCCI index. The new index can be applied to other similar urban tourist destinations and assist impact assessment studies and tourism management measures, including climate change adaptation.

**Keywords:** urban tourism; climate; climate change; tourist comfort indices; empirical survey

## 1. Introduction

Climate and weather are inextricably linked with various aspects of human life and are widely known to influence social attitude, daily schedules, traveling, recreational and outdoor activities, and mood. Tourism, largely depending on available climatic resources, is, therefore, a sector that is greatly affected by climatic factors and climate change [1–4].

Travel and tourism are highly prosperous and ever-expanding sectors of the global economy, constituting for many countries the primary source of income. The year 2018 was the ninth in a row of sustained growth for global tourism, with a 5% increase in international tourist arrivals from the previous year [5]. In Europe, hosting 51% of the world's international arrivals and 40% of international tourism receipts, Southern Europe and the Mediterranean region (mainly Italy, Greece, Portugal, and Croatia) witnessed an impressive 8% increase in tourist arrivals.

Meanwhile, urban destinations account for 44% of global international travel according to the "City Travel and Tourism Impact Report 2019" compiled by the World Travel and Tourism Council (London, UK). Although travel is differentiated from tourism, it can be a clear and useful indication of trends in the latter sector. Indeed, this report shows that 73 popular cities worldwide collectively contribute almost one quarter (24%) of the sector's world direct GDP and support over 17 million jobs in the sector. These figures clearly illustrate that urban tourism is a key component in city development and the urban economy.

It is now widely acknowledged that climate change risks (i.e., heat stress, inland and coastal flooding from rising sea levels, storm surges and extreme precipitation, landslides, and water scarcity from droughts and increased aridity) in several urban areas are increasing, with widespread adverse effects on human health and livelihoods, local and national economies, and ecosystems [6–9]. Despite these significant findings, it has been documented across several international reviews of tourism and climate change studies that very limited attention is given to urban tourism [10–16]. The existing variety of approaches that address the issue of climate change's impact on tourism focus almost exclusively on the beach and ski tourism industries. Traditionally, a widely applied methodology for assessing climate change's impact on tourism has been the use of the so-called climate indices, which provide a measure of the suitability of a location in terms of its climatic conditions for tourism. They can also be used together with climate projections to provide an estimation of climate change's future impact on the tourism industry.

A plethora of climate indices have been developed, each attempting to expand upon the versatility and function of the previous ones [17–27]. However, although there have been many applications of climate indices to various cities [17–23], so far none of these indices has been informed by empirical data collected exclusively at urban tourist destinations. Considering that the climatic preferences of tourists visiting urban areas may differ from those of tourists visiting the countryside or beach resorts, it is uncertain whether existing climate comfort indices are suitable for use in urban tourism. The goal of the present work is to cover this gap by developing a new index, the Urban Climate Comfort Index (UCCI), which integrates all climate variables that are meaningful for urban tourism and is informed by empirical data from an urban tourist destination. To this end, a core element of our study is an in-situ survey through questionnaires filled in by urban tourists to collect these data, allowing us to capture their actual climatic preferences and perceptions. We further decided to focus our study on Mediterranean urban areas, which are characterized by dry and hot summers and attract a very large number of tourists every year. Athens, located at the European part of the Mediterranean coast, was selected as a representative example of such urban tourist destinations. It hosts, among numerous other touristic sites, the Acropolis, which is one of the most visited tourist attractions in the world, making Athens particularly suitable for investigating the climatic preferences of urban tourists. The questionnaires used in the present study build on the content of questionnaires used in previous research on the climate dimensions of other forms of tourism [23].

After reviewing the available literature on the relationship between climate and tourist preferences in Section 2, the paper presents in Section 3 the empirical study and the methodology that was followed for each separate climatic variable examined. The key findings of the empirical study are presented in Section 4 along with the final formulation of the UCCI index. Finally, Section 5 discusses key uncertainties and limitations of the study and highlights the areas for further research.

## 2. Literature Review

A climate index is a quantitative measure for describing the suitability of regional/local climatic characteristics for tourism-based recreational activities and, thus, can be very useful in the decision making of tourists and travel agents alike. As already mentioned in Section 1, climate indices may also be used in conjunction with future climate projections to allow an estimation of future climate change risks for the tourism industry. This can facilitate and guide the formulation and implementation of policies for climate change adaptation in the tourism industry. During the past 30 years, many studies have attempted to develop new climate indices or improve existing ones. However, many studies carried out in the context of outdoor spaces in an urban environment [28–31] focused only on the thermal component of bioclimatic comfort (i.e., temperature, relative humidity, radiative exchanges, etc.) for permanent residents. Thus, they did not provide enough insight on how tourists, who often are unfamiliar with local climatic conditions and spend a short time at a destination,

perceive the overall climatic comfort of this location, which also includes more facets than the thermal-related factors [4].

Each climate index utilizes slightly different climate variables, weighting schemes, and approaches in computing the scores of each variable and the overall score. A well-known climate index, being the first developed for tourism, is the Tourism Climate Index (TCI) [24]. This index provides a score in a percentage form, ranging from 0% to 100%, with 0% corresponding to unacceptable climate conditions and 100% representing ideal climate conditions. The TCI has been applied to various types of tourism, populations, and environments [32–37]. In two separate studies conducted in Australia [34] and Europe [37], the TCI was utilized along with future climate projections under different global warming scenarios to discover the implications of climate change on tourism. It was found that climatic suitability for tourism in Australia is expected to deteriorate in northern urban locations and improve in southern locations, whereas opposite trends were estimated for Europe.

The TCI has been repeatedly criticized due to, inter alia, its weighting scheme, which is not informed by empirical studies and neglects the potential overriding effects of rain and wind, and its coarse time scale [20,25,38]. For example, the TCI assigns a weight of 40% to the thermal comfort, which, according to later research, was found to be too high [39]. Bearing the TCI's limitations in mind, de Freitas et al. [25] developed the Climate Index for Tourism (CIT), which integrates all facets of climate, i.e., thermal, physical, and aesthetic, into one single index and uses the concept of climate thresholds [40,41]. Following this, the Universal Thermal Climate Index (UTCI) [26] was originally designed to assess the thermo-physiological effects of the atmospheric environment on the human body and has been used in numerous studies since then [27,42–46]. Bleta et al. [27] used both the UTCI and PET (Physiological Equivalent Temperature) indices to assess the bioclimatic conditions of the island of Crete, showing, inter alia, that no distinct trend exists for the annual number of days with mean daily extreme values of PET/UTCI. In another study conducted on the Caribbean islands, Rutty and Scott [43] calculated the UTCI index for coastal tourism with concurrent micro-meteorological measurements and thermal perception surveys. They concluded that acceptable thermal conditions for beach tourism differ significantly from other tourism environments (i.e., urban, mountain, etc.), while personal characteristics and the climatic region of origin of tourists also influence perceptions on the suitability of thermal conditions for tourism activities. A similar study was conducted on the Baltic coasts of Poland by Kolendowicz et al. [42].

Recently, Scott et al. developed the Holiday Climate Index (HCI) and compared it with the TCI [20]. The HCI was developed for leisure tourism in urban and beach environments. A new feature of the HCI compared to the TCI was that its variable rating scales and weighting system were based on the recent literature on tourists' climatic preferences that were obtained from surveys in different countries around the world (but not focusing on urban tourist destinations). The index also made use of de Freitas's CIT index, overriding the effect of the physical parameters (wind, rain). The HCI generates a score ranging from "dangerous" (0–9) to "ideal" (90–100) climate conditions for tourism. Scott et al. used the HCI to rate the present and future climatic conditions in six major urban destinations in Europe (London, Paris, Barcelona, Rome, Istanbul, and Stockholm). The results, when compared to the TCI, showed consistently higher ratings all year round, with more frequent ideal conditions for urban tourism. This difference can be explained though the growing percentage of research that demonstrates that physical parameters such as rain are considered more influential for tourist satisfaction compared to the thermal component. Another possible reason for such discrepancies between the two indices is that people using outdoor spaces for recreational purposes may tolerate thermal conditions that exceed normal indoor comfort thresholds [20,29,47,48].

The realization that actual thermal and climatic preferences of tourists differ from those that are deduced from the thermo-physiological processes of the human body (due to adaptation, mitigation, and psychological measures) has led to an alternative approach to

the subject of climate comfort. During the past 20 years, an increasing body of research has focused on the documentation of tourists' climate perceptions and preferences by using questionnaires either ex situ or in-situ [22,49–56]. Many of these studies use the concept of thresholds for "ideal" and "unacceptable" climate conditions. Most of these studies focus on beach and coastal tourism, and only some of them address tourism in urban and mountain environments.

Scott et al. performed an ex-situ survey study of university students in Canada, New Zealand, and Sweden [50], investigating the tourists' preferences for four key climate variables, namely, air temperature, rain, wind, and cloudiness, and for three tourism environments (i.e., beach, urban, and mountainous). They found significant variations between the three environments, indicating again the environment dependency of optimal climate comfort conditions. Along this line, another study [57] formulated questionnaires for beach and urban tourism that were then administered to university students. Five urban Mediterranean locations were assessed, and, overall, a temperature range of 20–26 °C, a 15 min/day rain duration, 25% cloudiness, and a light breeze were found to be the ideal climate conditions for urban tourism during summer. Using a survey of Belgian and Dutch tourists who were to depart for Mediterranean destinations, Moreno (2010) found that the absence of rain was found to be the most important variable for summer beach tourism, followed by a comfortable temperature [51]. The importance of the absence of rain for beach tourism was confirmed by the research of Georgopoulou et al. (2018) [23] who developed the Beach Utility Index (BUI) based on an empirical study carried out in situ on Greek islands [23]. Finally, during an ex-situ survey of the French population, Dubois et al. (2016) concluded that the findings of similar studies are not necessarily replicable to other populations, destinations, and tourism environments. This study found that French coastal tourists were less tolerant to heat and more tolerant to cold, while it confirmed that the respondents found rain to be repulsive for coastal tourism-related activities [39].

Other applications of climate comfort schemes and indices for tourism have been developed, such as the CTIS [17], the MCIT [21], and the MOCI [19,58], to name a few. Most of the existing indices, however, focus on beach and coastal tourism rather than on urban tourism, while the indices assessing the climatic conditions in urban environments concentrate mainly on the thermal component of climate comfort. The characteristics of the various climatic indices developed so far are summarized in Table 1. As noted in Section 1, the present study aims to address the integrated climate comfort of urban tourists by drawing its conclusions from an in-situ survey in the city of Athens, Greece. A new Urban Climate Comfort Index (henceforth, UCCI) has been developed, established exclusively on the urban tourists' stated preferences, which allows us to better understand and possibly predict—when combined with climatic projections and tourist flow models—tourist flows during the hot and dry summers of Mediterranean climates.

**Table 1.** Overview of scientific literature on climate comfort indices and conditions for tourism.

| Index and Type of Tourism | Author (s) | Parameters | Weight Specification | Location of Application | Findings |
|---|---|---|---|---|---|
| TCI, general | Mieczkowski 1985 [24] | Daytime thermal comfort (T and RH) | Arbitrary | - | Development of TCI |
| TCI, general | Nicholls and Amelung (2008) [37] | | Arbitrary | Europe | TCI used with future climate projections. Climatic suitability for northern urban locations is likely to improve in the future, whereas for southern locations is projected to deteriorate |

**Table 1.** *Cont.*

| Index and Type of Tourism | Author (s) | Parameters | Weight Specification | Location of Application | Findings |
|---|---|---|---|---|---|
| TCI, general | Amelung and Nicholls (2014) [34] | | Arbitrary | Australia | TCI used with future climate projections. Climatic suitability for northern urban locations is likely to deteriorate in the future, whereas for southern locations is projected to improve |
| CIT, 3S tourism | De Freitas et al. (2008) [25] | Integrated body-atmosphere balance, cloudiness, wind and precipitation thresholds | Arbitrary, thresholds empirically validated through survey | - | Development of CIT |
| UTCI-PET, general tourism | Bleta et al. (2014) [27] | Ambient air temperature, relative humidity, water vapor pressure, solar radiation, and wind speed | Literature on physiological models of human thermoregulation | Crete island, Greece | No distinct trend exists for the annual number of days with mean daily extreme values of PET/UTCI |
| UTCI-PET, beach and urban tourism | Rutty and Scott (2014) [43] | Ambient air temperature, relative humidity, water vapor pressure, solar radiation, and wind speed | Literature on physiological models of human thermoregulation | Caribbean islands | Tourists preferred conditions that are up to 18 °C warmer than urban respondents in Lisbon (21–23 °C PET) and 10 °C warmer than in Taiwan (27–29 °C PET) |
| HCI, urban tourism | Scott et al. (2016) [20] | Daily max. T, mean RH, precipitation, wind, cloud cover | Literature on tourist's climatic preferences from surveys around the world | London (UK), Paris (France), Barcelona (Spain), Rome (Italy), Istanbul (Turkey), and Stockholm (Sweden) | The results, when compared to the TCI, showed consistently higher ratings all year round, with more frequent ideal conditions for urban tourism |
| BUI, Survey based on beach tourism | Georgopoulou et al. (2018) [23] | Temperature, precipitation, cloudiness, wind speed | Weights were based on tourist perceptions | Greek islands | Absence of rain most important variable for beach tourism. Development of BUI index based entirely on tourist weather preferences |
| Survey based on-beach, urban and mountain tourism | Scott et al. (2008) [50] | Temperature, precipitation, cloudiness, wind speed | The study only examines tourist weather preferences | Canada, New Zealand, Sweden | University student survey. Significant variations between the three environments. A median temperature of 22 °C, a 25% cloudiness level, and a light breeze (1–9 km/h) are the preferred conditions for urban tourism. Rain and temperature are the most important variables. |
| Survey based on beach and urban tourism | Rutty and Scott (2010) [57] | Temperature, precipitation, cloudiness, wind speed | The study only examines tourist weather preferences | Athens (Greece), Istanbul (Turkey), Marseilles (France), Barcelona (Spain), and Venice (Italy) | A temperature range of 20–26 °C, a 15 min/day rain duration, 25% cloudiness, and a light breeze are the ideal climate conditions for urban tourism during summer |

**Table 1.** *Cont.*

| Index and Type of Tourism | Author (s) | Parameters | Weight Specification | Location of Application | Findings |
|---|---|---|---|---|---|
| Survey based on beach tourism | Moreno (2010) [51] | Temperature, precipitation, cloudiness, wind speed | The study only examines tourist weather preferences | Survey of Belgian and Dutch tourists who were to depart for Mediterranean destinations | In this case, the respondents rated a mean temperature of 28 °C, a light breeze (1–9 km/h), and a 0% cloud cover to be the most suitable conditions for beach tourism. Absence of rain was the most important variable for summer beach tourism |
| Survey based on beach tourism | Dubois et al. (2016) [39] | Temperature, precipitation | The study only examines tourist weather preferences | Internet survey of French beach tourists | Beach tourists perceived heat generally as positive or neutral, and rain repulsive for tourism |

## 3. Materials and Methods

The Urban Climate Comfort Index (UCCI) of this study is based on the stated preferences of urban tourists in the Mediterranean capital city of Athens, Greece. A structured questionnaire was distributed to tourists visiting the Acropolis Museum, which is located very close to the Acropolis, a well-renowned national heritage monument. The place of the survey was specifically chosen since the Acropolis is located at the heart of the city center that hosts most of the accommodation units and cultural activities. The Acropolis Museum is a major tourist attraction, having welcomed more than 14.5 million tourists from all over the world since its grand opening in 2009 [59].

An initial questionnaire was first developed and distributed to a small group of tourists to test coherence, clarity, and time needed for answering all the questions. Next, we modified the questionnaire in order to make it shorter and clearer, and its final form was administered to random domestic and foreign tourists at the entrance of the museum. The simple random sampling method was chosen as it was determined to provide an unbiased representation of the studied group. An important aspect of the survey is that each questionnaire included a table with the values of the previously mentioned weather variables at the time of answering. Hence, the tourist could more easily assess and quantify his/her climate perception and provide more accurate answers. The questionnaire comprised four sections with seventeen questions.

The first section focused on demographics and tourist background information (i.e., nationality, country of residence, gender, age, education level).

The second section, named Travel experience in Greece, contained questions on the reasons for choosing Athens as a summer urban destination as well as the importance of several parameters possibly affecting their choice.

The perceptions on weather and summer tourism were the subject of the third section of the questionnaire. This section constituted the core element on which the UCCI index has been built. Tourists were asked to rate the importance of several climatic parameters for urban tourism, namely, comfortable temperature, absence of rain, absence of strong winds, absence of clouds, and comfortable level of humidity. Moreover, in the same section, tourists were asked to state the 'ideal' and 'unacceptable' values (thresholds) for each of the above-mentioned climatic parameters.

Finally, the fourth section focused on the reaction and potential adaptation measures to be applied by tourists in case of 'unacceptable' weather conditions during their summer holiday.

The survey was carried out both during weekdays and weekends, in late summer 2019. A total of 250 questionnaires were collected at the end of the survey, with a mean response rate of 94% as 16 questionnaires were deemed as either erroneous or incomplete and were therefore discarded from analysis. The survey was anonymous and no personal information was collected from the respondents.

### 3.1. Estimating Climate Comfort in an Urban Environment

The multifaceted nature of weather and the complex way climatic variables interact to form a tourism experience calls for a tool that is easily understood and integrates the effects of different climatic variables. This can be achieved with survey-based climate indices that can also be compared with other indices to test for inconsistencies between different schemes used (e.g., weights, climate variables chosen), ethnic groups, tourism environments, etc. Climate thresholds in particular offer a convenient tool to tourists when assessing climatic conditions.

In the present study, tourists were invited to characterize prescribed values (or range of values) of several climate variables such as ideal or unacceptable for urban tourism during summer. The resulting threshold values (or range of values) of a variable were then 'translated' into a 0–100 utility score distribution, which describes the preferences of tourists for that variable. This procedure, described in detail in Section 3.2, was followed for all climate variables, which were then weight-averaged (according to the weights assigned to each variable—see Section 3.3) to build the UCCI index.

A critical issue one experiences when developing a climate index is which climatic variables to include and what weight to assign to each of them. As already mentioned, most other climate indices used in the past for urban environments focused on the thermal aspect of climate comfort. Therefore, those indices consisted of a temperature-related variable and/or heat flux variable between the environment and the human body. However, tourist satisfaction is affected by several other factors that may be more influential compared to the thermal component, as recent studies have revealed. Following previous research [25,60], the current index was designed to include the thermal, physical, and aesthetic factors and the climate variables associated with the latter. Therefore, the climate variables considered here were the air temperature and relative humidity (thermal), rainfall (physical), wind speed (physical), and cloudiness (aesthetic).

### 3.2. Utility Functions

The aggregated tourists' preferences are expressed in the form of utility functions. As mentioned above, a utility score ranging from 0–100 is considered for each climatic variable, with 0 and 100 scores corresponding to minimum and maximum climate comfort for urban tourist activities, respectively. By answering specific questions in the questionnaire for each climate variable, each tourist identifies his/her ideal and unacceptable value (or range of values) for this variable. Ideal values are assigned a utility value of 100, while the unacceptable values are assigned a value of 0. The question then arises of what utility scores should one assign to the intermediate values between his/her ideal and unacceptable scores, or in other words what the shape of the utility function distribution curve of this climate variable for a particular respondent is. To properly determine this, additional questions on those intermediate values would be needed, but these take up too much of the time to fill in the questionnaire during the in-situ survey. Thus, it was assumed that at intermediate values of this climate variable, the utility scores for this respondent change linearly between 0 and 100.

Once this process for all respondents is completed, each distinct level of a specific climate variable ends up with different utility scores (as many as the total number of respondents). Then, for each distinct value of each climate variable, these scores are averaged, resulting in a composite utility score for each value and, hence, to a composite utility distribution for this climate variable. This utility distribution can then be mathematically expressed by an empirical utility function. The aggregation of utility functions for all climate variables into an overall utility function (i.e., the UCCI index) is performed by using the weights of the different climatic variables, which are also estimated empirically through the answers of urban tourists to the relevant questions posed in the context of our survey. The following Sections 3.2.1–3.2.5 describe in detail the estimation of the utility function for each climatic variable, while the weights' determination and the aggregation of the different utility functions into the UCCI index are presented in Sections 3.3 and 3.4, respectively.

### 3.2.1. Temperature

Each respondent is asked to locate the ideal, unacceptably cool, and unacceptably hot temperature value or range of values ($T_{ideal}$, $T_{cold}$, $T_{hot}$) for urban summer tourism at an interval from 11 °C to 40 °C, with a 1 °C interval step. The temperature values rated as 'ideal' are assigned a maximum utility score of 100, while those rated as 'unacceptably cold' and 'unacceptably hot' are assigned a minimum utility score of 0. Assuming a linear evolution between the maximum and minimum utility, the utility score of an intermediate temperature value ($T_n$) is given by

$$TU_{m,n} = 100 \frac{T_n - Tcold_m}{Tideal_m - Tcold_m} \text{ for } Tcold_m \leq T_n \leq Tideal_m \quad (1)$$

$$TU_{m,n} = 100 \frac{T_n - Thot_m}{Tideal_m - Thot_m} \text{ for } Tideal_m \leq T_n \leq Thot_m \quad (2)$$

$$TU_{m,n} = 0 \text{ for } T_n \leq Tcold_m \vee T_n > Thot_m \quad (3)$$

where $m$: the number of respondents and $T_n$: temperature value ranging from 11 °C to 40 °C.

Hence, for each respondent, a temperature utility profile is computed. Next, by computing the average utility score for each temperature value using the utility scores for this specific value from all respondents, a composite temperature utility score profile is generated that reflects the temperature preferences of all the respondents to the survey

$$TU_n = \frac{\sum_{m=1}^{m=N} TU_{m,n}}{N} \quad (4)$$

Following this, the composite temperature utility profile is normalized so that the highest utility score is scaled to a value of 100. The remaining values are also calibrated, respectively. Thus

$$TUideal\_scaled = 100 = max(TU_n)$$

$$\text{thus } TUscaled_n = \frac{TU_n}{max(TU_n)} 100 \quad (5)$$

### 3.2.2. Relative Humidity

Relative humidity (RH) has been included in the UCCI index since it indirectly affects thermal comfort levels by interfering with the thermo-regulatory mechanisms of the body. Since warmer air has a higher moisture holding capacity compared to cooler air, a property dictated by the Clausius–Clapeyron equation, warm and humid air during summer hinders the effective evaporation of sweat and therefore heat loss. On the other hand, very low levels of relative humidity have significant negative effects on the eyes, skin, and mucous membranes, thereby, potentially affecting tourist health and comfort [7].

The participants are asked to indicate their ideal and unacceptable relative humidity values (or range of values) for their summer holidays in a city. The available answers ranged from 0% to 100%, with an interval step of 10%. The participants are also informed about the relative humidity at the time of the survey. In addition, a note was included in the questionnaire describing extreme values of RH and where these occur naturally in the world. Specifically, RH values close to 0 can occur in arid and desert-like environments, whereas RH values close to 100% usually occur in tropical regions (e.g., tropical coastal and forested areas). This information was included in the questionnaire as it has long been acknowledged that humans usually are not able to adequately estimate relative humidity [61–63]. For all these reasons, the assumption is made that both values of 0% and 100% (boundaries) are also unacceptable (zero utility score) unless stated otherwise.

The utility scores for an intermediate value of RH ($H_n$) and for the extreme (boundary) RH values of 0% and 100% are given by the following equations

$$HU_{m,n} = 100 \frac{H_n - Hunac_m}{Hideal_m - Hunac_m} \tag{6}$$

for $Hideal_m \leq H_n \leq Hunac_m$ or $Hideal_m \geq H_n \leq Hunac_m$

$$HU_{m,RH=0} = 0 \; \kappa\alpha\iota \; HU_{m,RH=100} = 0 \text{ (boundary values)} \tag{7}$$

$$HU_{m,n} = 0 \tag{8}$$

for RH $\neq$ 100 and $H_n > Hunac_m$ or RH $\neq$ 0 and $H_n < Hunac_m$
where $m$: the number of respondent and $H_n$: RH value ranging from 0% to 100%.

The utility profiles from all respondents are combined into one 'composite' relative humidity utility profile. As with ambient air temperature, this is achieved by computing the average from all respondents' utility scores for each relative humidity percentage value

$$HU_n = \frac{\sum_{m=1}^{m=N} HU_{m,n}}{N} \tag{9}$$

Finally, a normalization is performed so that the composite utility scores are scaled to a score range of 0–100

$$HUideal\_scaled = 100 = max(HU_n) \text{ and } HUscaled_n = \frac{HU_n}{max(HU_n)} 100 \tag{10}$$

### 3.2.3. Rainfall

The participants are asked to rate their rainfall preference by indicating the ideal and 'unacceptable' rainfall duration ($Sideal_m$, $Sunac_m$) in minutes (or hours) per day during their vacations at the urban destination. Each respondent is presented with rainfall duration values ranging from 0 min to 5 h, with various interval steps. As before, a utility value of 100 is assigned to the ideal rainfall duration and a zero utility value to the unacceptable rainfall duration. Utility at all intermediate rainfall duration values is assumed to change in a linear way. Furthermore, it is assumed that any rainfall duration smaller than the stated ideal one corresponds also to a score of 100 (i.e., is also ideal) unless stated otherwise (notably, a few respondents replied that a 0 min rainfall is unacceptable for them), as in several research studies rainfall has been consistently found to reduce tourist satisfaction. Similarly, any rainfall duration higher than the stated unacceptable level is assumed to correspond to a zero utility score. The equations for calculating the utility scores for an intermediate rainfall duration Rn are given below

$$RU_{m,n} = 100 \frac{R_n - Runac_m}{Rideal_m - Runac_m} \text{ for } Rideal_m \leq R_n \leq Runac_m \tag{11}$$

$$RU_{m,n} = 0 \text{ for } R_n > Runac_m \tag{12}$$

$$RU_{m,n} = 100 \text{ for } R_n < Rideal_m \tag{13}$$

where $m$: the number of respondent and $R_n$: rainfall duration value ranging from 0 min to 5 h.

The profiles from all respondents are then combined into one composite rainfall utility profile. Again, this is accomplished by computing the average of all respondents' utility scores for each rainfall duration value

$$RU_n = \frac{\sum_{m=1}^{m=N} SU_{m,n}}{N} \tag{14}$$

A normalization is performed so that the utility scores are scaled to a score range of 0–100

$$RUideal\_scaled = 100 = max(RU_n) \text{ and}$$

$$RUscaled_n = \frac{RU_n}{max(RU_n)}100 \tag{15}$$

### 3.2.4. Cloudiness

Respondents are invited to rate their preferred sky condition by indicating the ideal and unacceptable cloud coverage ($Cideal_m$, $Cunac_m$). The cloud coverage is presented to each participant in the survey in the form of percentages, with 0% indicating a clear sky and 100% indicating a complete overcast sky. The available choices are given in 10% interval steps and participants who do not indicate an unacceptable level of cloudiness are excluded from the calculation (since a 0 utility cannot be calculated for them). Once again, for each respondent, the utility scores for intermediate cloudiness values $C_n$ between the stated as ideal (100%) and unacceptable (0%) are determined in a linear way as in the previous variables

$$CU_{m,n} = 100\frac{C_n - Cunac_m}{Cideal_m - Cunac_m} \text{ for } Cideal_m \leq C_n \leq Cunac_m \tag{16}$$

$$CU_{m,n} = 0 \text{ for } C_n > Cunac_m \tag{17}$$

$$CU_{m,n} = 100 \text{ for } C_n < Cideal_m \tag{18}$$

where $m$: the number of respondent and $C_n$: cloudiness value ranging from 0% to 100%.

Again, a composite cloudiness utility profile is created by calculating for each cloudiness level the average utility score value of all respondents' score values

$$CU_n = \frac{\sum_{m=1}^{m=N} CU_{m,n}}{N} \tag{19}$$

A normalization is performed to scale the results to a 0–100 magnitude scale, where the highest utility score is assigned to the value of 100 and the rest of the levels are adjusted accordingly:

$$CUideal\_scaled = 100 = max(CU_n) \text{ and } CUscaled_n = \frac{CU_n}{max(CU_n)}100 \tag{20}$$

### 3.2.5. Wind

The participants to the survey are asked to state their preference in terms of windiness during their summer vacations at an urban destination. They are presented with different choices in both a descriptive and numerical form (Table 2, i.e., 'no wind', 'light breeze' (1–2 Beaufort or 1–7 mph), 'moderate wind' (3–5 Beaufort or 8–24 mph), 'strong wind' (6–7 Beaufort or 25–38 mph), and 'very strong wind' (8–9 Beaufort or 39–54 mph). The numerical values are given in ranges since it was deemed difficult for tourists to indicate a specific wind speed value in specifying their preferences.

**Table 2.** Wind profiles of the current study associated with corresponding Beaufort scale values.

| Wind Profile | Beaufort Scale | Wind Speed (m/s) |
|---|---|---|
| No wind | Calm (0 bft) | <0.5 |
| Light breeze | Light air–light breeze (1–2 bft) | 0.5–3.3 |
| Moderate wind | Gentle breeze–fresh breeze (3–5 bft) | 3.4–10.7 |
| Strong wind | Strong breeze–near gale (6–7 bft) | 10.8–17.1 |
| Very strong wind | Gale–severe gale (8–9 bft) | 17.2–24.4 |

In the case of wind for urban tourism, the utility function is based on divergence of the views expressed [23]. In that sense, the 'theoretically ideal' wind profile would be the one that is evaluated as ideal by all respondents and by none of them as unacceptable. Likewise, the 'theoretically unacceptable' wind profile would be the one that is evaluated as unacceptable by all respondents and by none of them as ideal. In an x-y graph where the x-axis stands for the % of respondents who find a specific wind profile as ideal ($I_n$) and the y-axis stands for the % of respondents who consider it as unacceptable ($U_n$), the distance $d_{worst}$ between the 'theoretically unacceptable' wind profile (i.e., the 'worst' profile) and the 'theoretically ideal' wind profile is

$$d_{worst} = \sqrt{100 + 100} = 141.4 \tag{21}$$

and the distance $d_n$ between a specific wind profile as defined in this survey (i.e., 'no wind', 'light breeze', 'moderate wind', 'strong wind', or 'very strong wind') and the 'theoretically ideal' wind profile is

$$d_n = \sqrt{(1 - I_n)100 + U_n 100} \tag{22}$$

Since the $d_{worst}$ corresponds to a utility equal to 0, the utility score for a distinct wind profile with a distance of $d_n$ from the 'theoretically ideal' is given by

$$WU_n = 100 - 100 \frac{d_n}{d_{worst}} \tag{23}$$

Finally, the calculated utility scores are normalized, as for the previous variables, to a scale of 0–100

$$WUideal\_scaled = 100 = max(WU_n) \text{ and } WUscaled_n = \frac{WU_n}{max(WU_n)} 100 \tag{24}$$

To develop a continuous utility function that can be applied for any wind velocity, the actual numerical wind speed values of the Beaufort scale assigned to each wind profile can be utilized, assuming that for a wind speed within the range of each of the five distinct wind profiles of Table 2 the utility score remains constant to the one calculated by Equation (24).

### 3.3. Climate Variables' Weighting Scheme

The weights of the selected climate variables in the UCCI index are also empirically determined from the findings of the survey. To this end, question 2.3 of the questionnaire requests the tourist to mention up to three reasons for selecting Greece for his/her urban summer vacations. Following this, question 2.4 examines the importance (on a scale of 0 to 5, with 5 being the most important) of several pre-selected parameters (including 'Climate') for choosing Athens as a summer urban tourist destination. Finally, question 2.5 investigates the importance of the different climate variables used for building the UCCI index. The respondents can answer on a scale from 0 to 5 (as in question 2.4), with 5 indicating the highest importance.

Since our intention is to calculate the weights of the different climate variables based on the answers of urban tourists whose satisfaction is greatly influenced by weather elements, while some tourists may choose Athens for non-climate related reasons, the latter responses are excluded from the weight calculation process. Therefore, questions 2.3 and 2.4 aim to filter-in respondents. Specifically, responses to question 2.3 that use words such as 'weather', 'beaches', 'climate', 'sunny', 'warm', 'sailing', 'sea' etc. and/or assign high importance on 'Climate' in question 2.4 are considered suitable for inclusion in the calculation of weights.

### 3.4. Urban Climate Comfort Index (UCCI)

Once the utility score profiles for the different climate variables have been determined, the mathematical function (i.e., the utility theoretical distribution) that best describes a given profile can be determined as well.

After having determined both the weights and the corresponding utility functions, the following formula gives the UCCI score

$$UCCI_m = w_T TU_m + w_W WU_m + w_C CU_m + w_R RU_m + w_{RH} RHU_m \tag{25}$$

where $w_T$, $w_W$, $w_C$, $w_R$, $w_{RH}$ are the respective weights of temperature, wind, cloudiness, rainfall, and relative humidity, and $TU_m$, $WU_m$, $CU_m$, $RU_m$, $HU_m$ their corresponding utility functions.

As the UCCI aims to measure the climate suitability of a given urban destination for tourist activities, a classification scheme needs to be developed, 'translating' UCCI scores into suitability scores. To this end, a qualitative classification scheme such as that of Georgopoulou et al. [23] is proposed here, which comprises five potential suitability classes (Table 3):

**Table 3.** Classification of climate suitability for an urban tourist destination based on UCCI.

| UCCI Score | Climate Suitability |
| --- | --- |
| 80–100 | Excellent |
| 70–79 | Very good |
| 60–69 | Good |
| 40–59 | Acceptable |
| 30–39 | Unfavorable |
| 20–29 | Very unfavorable |
| 0–19 | Extremely unfavorable |

## 4. Results

### 4.1. Demographics of Respondents

The first section of the questionnaire covers background information on the respondent, namely nationality, country of residence, gender, age, and level of education. In total, 250 'acceptable' questionnaires were collected, all of which comprised answers to all questions of this first section apart from that on educational level, which was answered by 97% of the respondents.

Most of the respondents (61.2%) were women. This gender imbalance may have introduced a gender-based bias, although previous studies have not indicated any significant difference in climate perception and preferences between genders.

Regarding the age of individuals, 8.8% of respondents were less than 18 years old, 24.4% were in their 20 s, 25.2% in their 30 s, 15.6% in their 40 s, 14.4% in their 50 s, and 11.6% were over 61 years old (Figure 1). This indicates a typical age distribution of tourist flows, as the most active ages between 18 and 40 years old represent nearly 50% of the current sample.

Figure 2 shows the nationality distribution of the sample (250 respondents). Many participants, namely one fifth of the sample (26%), live in the United States, while the second (10.8%) and third (8.4%) largest part of the sample lives in the UK and France, respectively. About 7% of the sample were Greek citizens. Therefore, no specific pattern in the country origin of the participants was observed. It is noted that although both questions on nationality and country of residence were included in the questionnaire, only the latter is presented here since it was considered that this demographic characteristic affects the climate perceptions and preferences of tourists more.

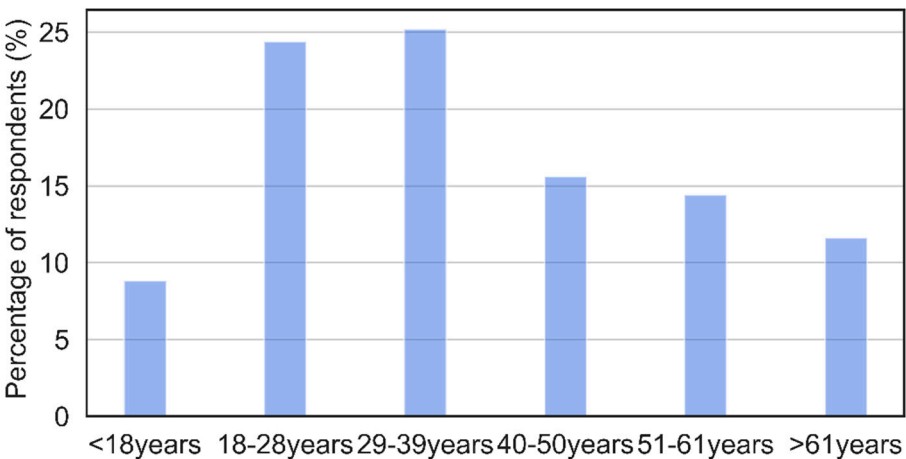

**Figure 1.** Age distribution of the respondents to the survey.

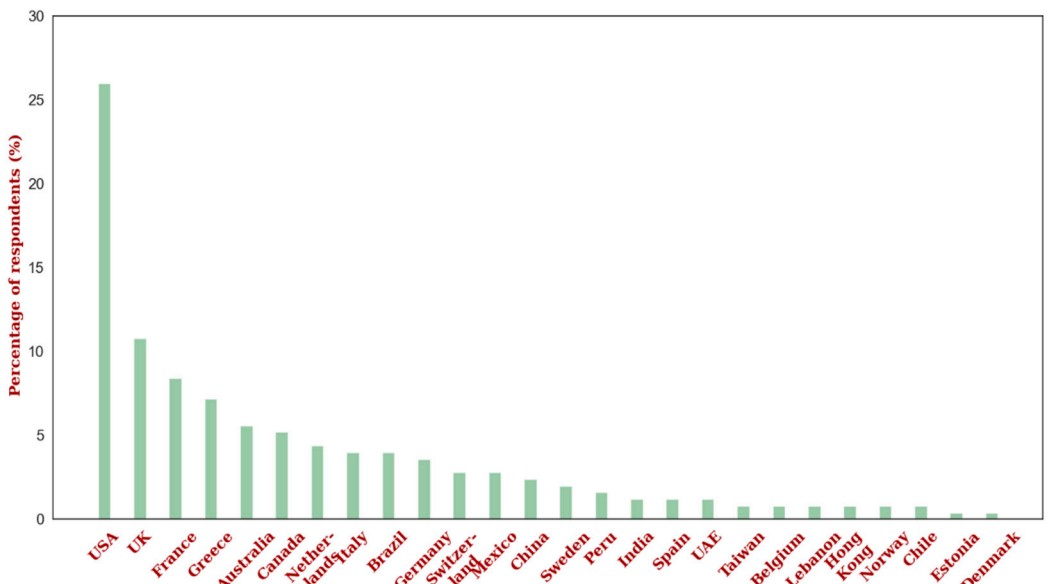

**Figure 2.** Countries of residence of the respondents to the survey.

Finally, regarding education level (Figure 3), a notable percentage of 81% of respondents had received higher education, with 43.8% having a university degree and another 37.2% a master's degree or higher. The rest of the respondents had completed a secondary or technical school. Thus, most tourists in our sample were highly educated.

### 4.2. Weights of Climatic Variables

As mentioned in Section 3.3, question 2.4 of the questionnaire examines the importance of several pre-selected parameters for choosing Athens as a summer urban tourist destination. For the determination of the weights of climatic variables, only those questionnaires that indicated 'climate' as one of the parameters were considered, as it was considered that these respondents had a real interest or concern in climatic conditions during their stay in the urban destination, and therefore their answers would provide more accurate results. A total of 145 questionnaires (58%) passed through the above filtering process, and their answers to question 2.5, investigating the importance of the five climate variables in the UCCI index, were utilized to compute the weights of climate variables.

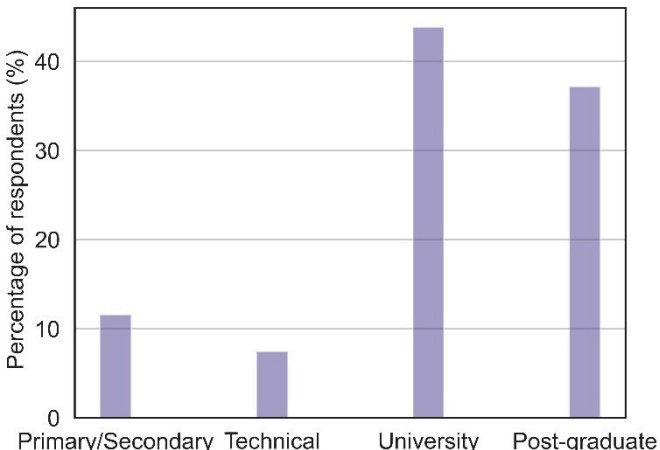

**Figure 3.** Education level of the respondents.

As said above, when evaluating the importance of the different climatic variables for urban tourism, the respondents could choose from a scale ranging from 0 to 5 (0: not important, 5: very important). On the assumption that the importance changes linearly along this qualitative scale, the importance scores of all 145 questionnaires for each climate variable were summed and averaged. Weights were then normalized on a scale from 0% to 100%.

The resulting weights of the different climatic variables (Figure 4 and Table 4) do not differ much, apart from that of the 'absence of clouds', which is the lowest (15.35%). Comfortable temperature was evaluated by the respondents as the most important for urban tourism, followed by the absence of rainfall. The weight values agree with Scott et al. (Scott et al., 2008) [50] who found similar results for urban tourism during summer. The relatively low weight of the absence of clouds is not a surprise since cloudiness is rather an aesthetic factor that does not impede tourist activities in urban destinations.

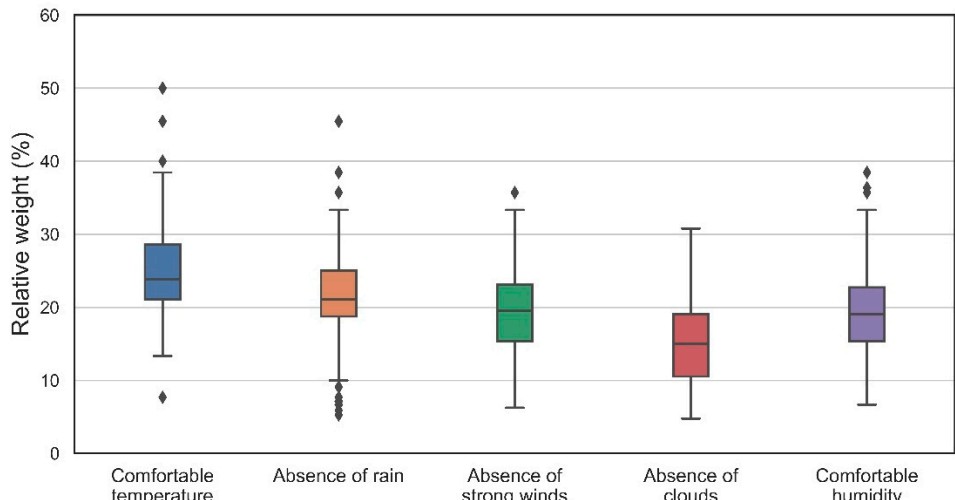

**Figure 4.** Weights of the different climate variables in the context of urban tourism, derived from the present survey. The whiskers indicate the maximum and minimum values, excluding the outliers. The horizontal black line indicates the median, while the range of the box height displays the upper and lower quartile. The rhombus points indicate the outliers.

**Table 4.** Weights of climate variable by nationality group in the context of urban tourism derived from the present survey.

|  | Comfortable Temperature | Absence of Rain | Absence of Strong Winds | Absence of Clouds | Comfortable Humidity |
|---|---|---|---|---|---|
| Northern Europe | 25.6% | 22.1% | 18.4% | 16.3% | 17.6% |
| Mediterranean | 23.1% | 22.5% | 20.1% | 15.5% | 18.7% |
| North America | 26.3% | 20.3% | 17.5% | 13.6% | 22.2% |
| South America | 24.3% | 23.3% | 19.6% | 14.8% | 17.9% |
| East Asia | 26.0% | 22.1% | 19.0% | 14.9% | 18.0% |
| Oceania | 21.7% | 22.9% | 21.5% | 12.4% | 21.4% |
| Overall | 24.5% | 21.8% | 19.1% | 15.3% | 19.3% |

The boxplot in Figure 4 illustrates the distribution of weights for each climate variable resulting from the respondents' answers. The distribution of urban tourists' views is similar for the different climatic variables, albeit somewhat narrower for ambient temperature and absence of rainfall, indicating that tourists agree more on the latter variables. However, it is noted that these two climate variables also present the most outliers compared to the rest.

The fact that the weights determined from our survey do not differ significantly is consistent with the findings of relevant studies (e.g., [50]). The latter also found that the selected climate parameters acquire lower absolute values of importance in urban tourism compared to other types of tourism as, in the former, the exposure of tourists to atmospheric conditions is lower. In other words, the urban environment is perceived by tourists to be more 'sheltered' against weather elements compared to coastal or mountainous environments.

To investigate any patterns of climatic preferences with respect to tourist nationalities, the respondents were divided in six different geographical groups, namely Northern Europe, Mediterranean, North America, South America, East Asia, and Oceania. The weights of the different climatic variables per nationality group are presented in Table 4 and Figure 5.

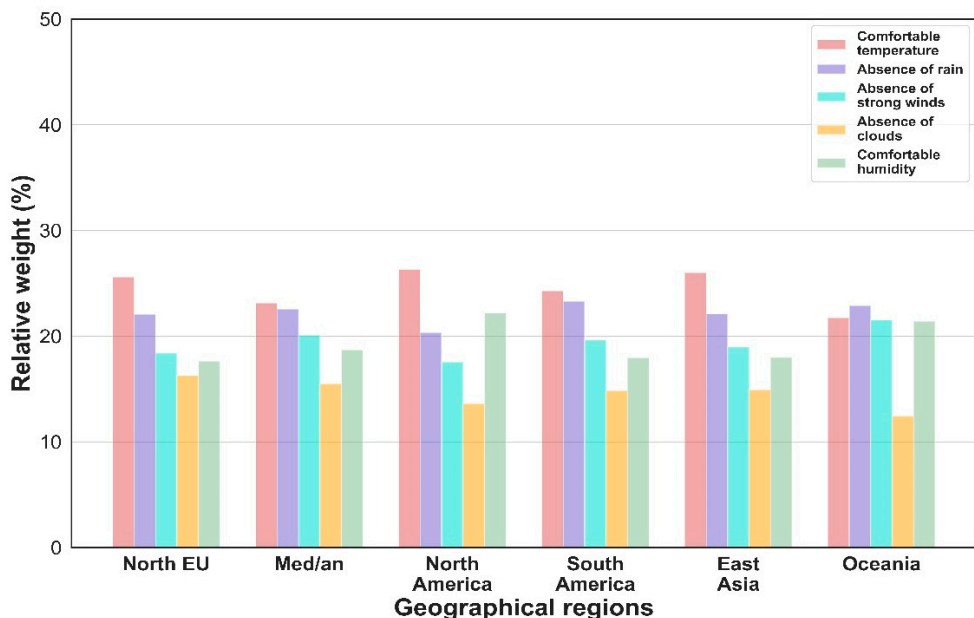

**Figure 5.** Weights of the different climate variables per nationality group.

As seen from Table 4 and Figure 5, no apparent deviations between the weights of nationality groups and those in the overall sample were found. For all nationality groups, comfortable temperature is the most important for urban tourism during summer, followed by the absence of rain. A comfortable humidity is considered as the least important for

all nationalities apart from Oceania and North America where its importance is close to that of comfortable temperature. In the case of Oceania, this is somewhat expected as it comprises regions with extreme values of relative humidity such as rainforests. However, the number of respondents in these two latter nationality groups is too small to draw definite conclusions.

### 4.3. Utility Functions of Climate Variables

#### 4.3.1. Ambient Air Temperature

The distribution of the temperature thresholds specified by the respondents regarding the ideal, unacceptably cool, and unacceptably hot ambient air temperature in the context of urban tourism is shown in Figure 6. The figure reveals that most of the respondents (58% or more) perceive a temperature of 15 °C as unacceptably cold for urban tourism during summer. Meanwhile, a temperature of 20 °C is perceived as ideal and too cold by equal percentages, indicating that values below this threshold are progressively considered as unacceptably cold for urban tourism. On the other hand, a temperature of 35 °C is perceived as unacceptably hot for urban tourism during summer by most respondents (63.5% or more). At the same time, a value of 32.3 °C was identified by an equal number of respondents as both ideal and too hot, indicating that temperatures higher than this threshold are progressively considered as unacceptably hot for urban tourism activities.

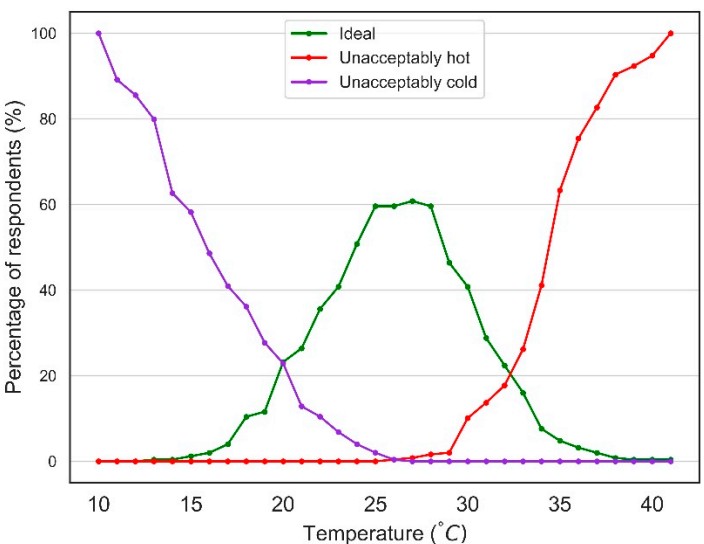

**Figure 6.** Percentage of respondents perceiving a specific temperature as ideal (green line), unacceptably hot (red line), and unacceptably cold (blue line).

A finding worth mentioning is that more than 58% of tourists prefer an ambient air temperature between 24 °C and 28 °C for their summer vacations in the city, which is relatively low compared to previous studies on tourism in beach or mountainous environments. For example, many past studies on beach tourism [4,51,57] found that an ideal preferred temperature lies in the range of 27–32 °C. Such differences are considered sound as the climatic preferences of urban tourists differ from those of beach tourists, with the latter found to tolerate higher ambient air temperatures (possibly because sea bathing offers them an effective way to adapt to heat).

Our findings agree with the results of other studies on thermal comfort conducted in urban environments such as that in the present study. Salata et al. [58], after surveying 1000 respondents in Rome, found a preferred temperature of 24.9 °C, while Scott et al. [50] identified 22.5 °C as ideal for urban vacations in New Zealand and Sweden. Another study carried out by Rutty and Scott [4], assessing thermal conditions for tourism in five major Mediterranean urban destinations (namely Athens, Istanbul, Marseilles, Barcelona, and

Venice), found that ideal air temperatures lie at 20–26 °C, while the unacceptably cold and unacceptably hot thresholds were 17 °C and 30 °C, respectively.

Following the approach described in Section 3.2 (Equations (1)–(4)), an empirical utility function for ambient air temperature was computed from the answers of all respondents. The function follows a Weibull distribution, with a shape parameter of c = 4.06 and a sigma value of σ = 20.1 (Figure 7). The utility function reaches a maximum score at 26.5 °C, while it has a coefficient of determination of $R^2$ = 0.98, which indicates a high degree of fit.

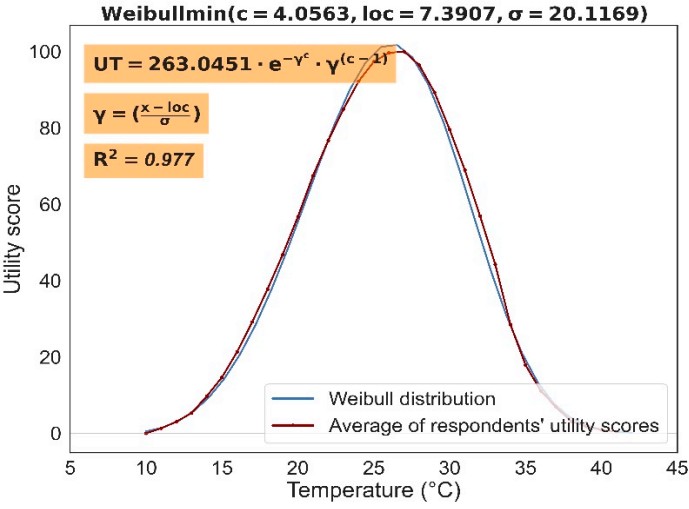

**Figure 7.** Empirical utility function for ambient air temperature in urban tourism.

The empirically estimated equation reflecting the utility function $TU_n$ for urban tourism associated with an ambient air temperature $T_n$ (in °C) is

$$TU_n = 263.0451e^{-(\gamma)^c} \cdot \gamma^{(c-1)} \; with \; \gamma = \frac{T_n - loc}{\sigma} \tag{26}$$

with the values of coefficients c, loc, and σ shown in Figure 7.

The utility function is valid for temperature values between 10 °C and 41 °C as this was the range explored in our questionnaire.

### 4.3.2. Rainfall

Figure 8 presents the percentage of respondents considering different values of rainfall duration as ideal and unacceptable. As shown, the great majority of respondents (about 90%) prefer a daily rainfall duration of 1 h or less for urban tourist activities, with 40% indicating zero hours of rain as ideal for urban vacations during summer. On the other end of the scale, nearly all respondents consider 5 h of rainfall during the day to be unacceptable and 62% of the respondents consider 3 h of rainfall to be unacceptable. It must be mentioned here that 25 and 35 respondents did not indicate an ideal and unacceptable rain duration, respectively, and, therefore, their responses to the questionnaire were excluded from further processing.

The above results agree with those of similar studies (e.g., [49]) who found that 57.2–67.2% of the reviewed tourists at tourist destinations in Catalonia consider summer tourism to be ruined if rain lasts more than 3 h. Notwithstanding this finding, this threshold value is larger compared to that found in beach tourism studies [4,39,57]. Most of the latter suggest that a rainfall duration of more than 2 h is unacceptable, while at the same time 0 h is considered ideal. This clearly indicates that urban tourists have a higher tolerance compared to beach tourists when it comes to rain, a reasonable conclusion as rain hinders beach tourists from going to the beach, whereas urban tourists can undertake additional activities indoors in the case of rainy weather (e.g., visit more museums, spend more time in shops, etc.).

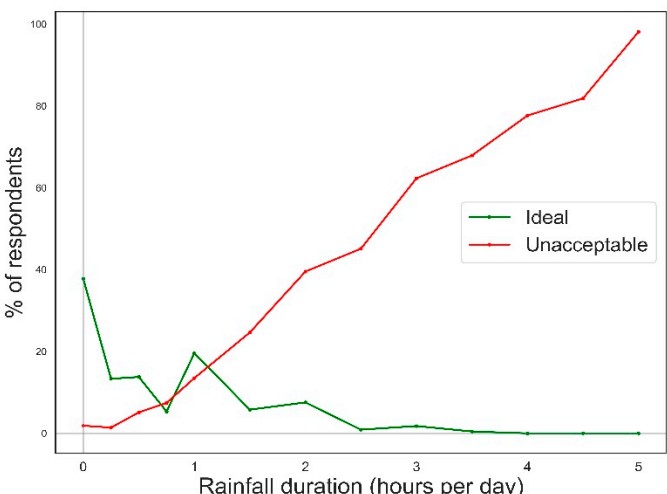

**Figure 8.** Percentage of respondents considering a rainfall duration as ideal (green line) and unacceptable (red line).

The rainfall utility function is based upon the previously mentioned Equations (11)–(13) and is shown in Figure 9.

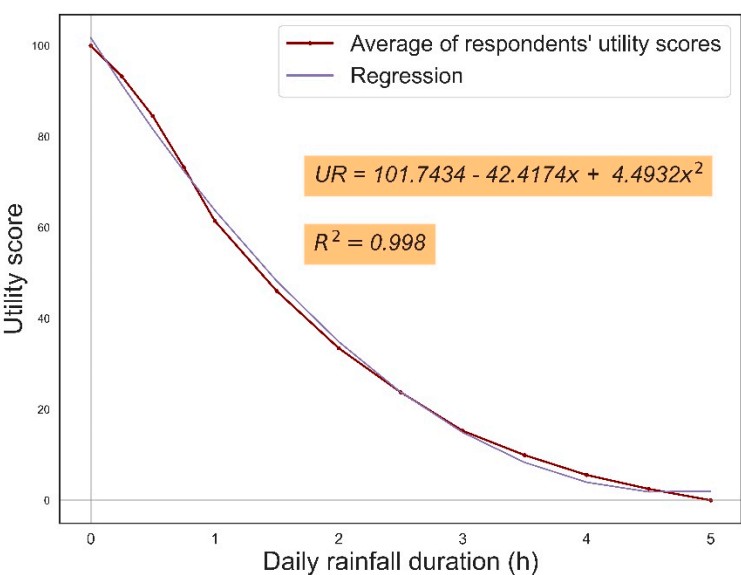

**Figure 9.** Empirical utility function for rainfall in urban tourism.

The equation of the fitted rainfall utility function is a second-degree polynomial with the following formula

$$UR = 101.7434 - 42.4174R_n + 4.4932R_n^2 \tag{27}$$

where $R_n$ is the rainfall duration (in h/day).

### 4.3.3. Wind

The wind utility function presents a more complex behavior. As said in Section 3.2.5, tourists were asked to indicate their preferred wind conditions during their urban vacations by choosing between five pre-established wind profiles. Figure 10 shows the percentage of respondents who consider each wind profile as ideal and unacceptable. The preferred wind condition for almost 60% of the respondents is a light breeze (1–2 Beaufort or 1–7 mph or 0.5–3 m/s), while nearly all respondents (98%) considered very strong winds (8–9 Beaufort or 39–54 mph) as unacceptable.

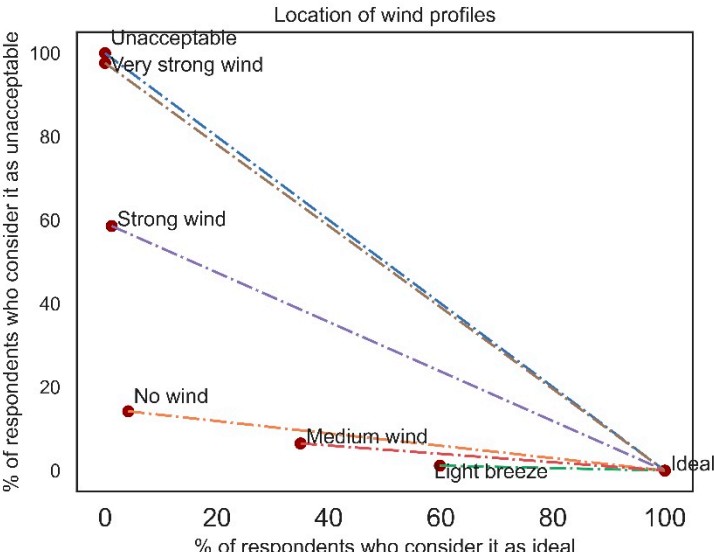

**Figure 10.** Percentage of respondents considering a specific wind profile as ideal and unacceptable, and relevant 'distance' between these and the 'theoretically ideal'.

Our results agree with those of Scott et al. [50] who identified a light breeze of 1–9 km/h (0.2–2.5 m/s) as ideal for urban tourism. In addition, the results of the current study agree with those for beach tourism, most of which indicate an ideal wind profile of 1–9 km/h (0.2–2.5 m/s) [4,49–51,57], etc.

The wind utility function is shown in Figure 11. The fitted wind utility function follows a fourth-degree polynomial, with a coefficient of determination of $R^2 = 0.98$. The utility function presents a wider shape rather than a sharp one, indicating that tourists exhibit a tolerance to high wind speed values up to about 14 m/s (after this point the utility score drops to less than 30%). It is noted that even a wind speed of about 11 m/s (i.e., a strong wind) receives a utility score of approximately 50%.

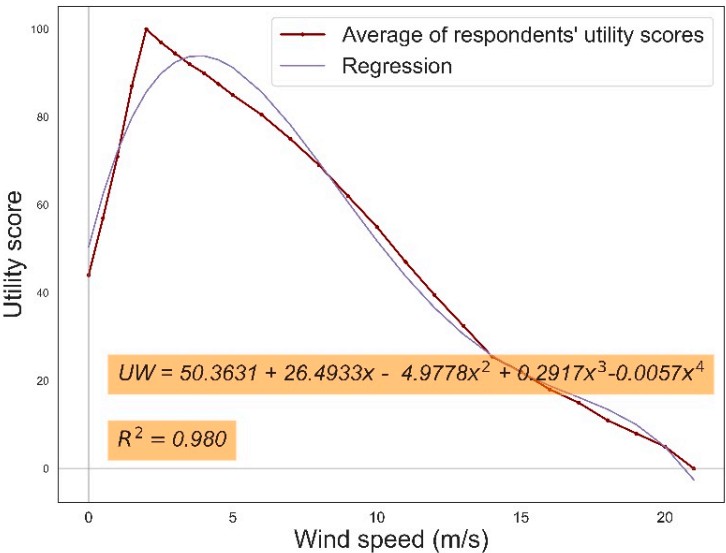

**Figure 11.** Empirical utility function for wind in urban tourism.

The mathematical formula of the polynomial is

$$UW = 50.3631 + 26.4933W_n - 4.9778W_n^2 + 0.2917W_n^3 - 0.0057W_n^4 \tag{28}$$

where $W_n$ is the wind speed value (in m/s) for which we wish to calculate the utility score.

### 4.3.4. Cloudiness

The results of the survey demonstrate a preferred cloud coverage of 25% or less for more than 95% of respondents. However, it must be mentioned that three respondents (out of the 250 participants) did not fill in the cloudiness section. From the remaining 247 respondents, 87 did not indicate an unacceptable cloudiness level and were therefore excluded from the calculation of the cloudiness utility function. The fact that 35% of total respondents did not indicate an unacceptable cloudiness level could illustrate the low significance that a high level of cloud coverage has for urban tourism. Of the other 160 respondents, 91.4% indicated a complete overcast sky (100% clouds) to be unacceptable, while 67.5% indicated a cloudiness of 90% or more as unacceptable for urban tourism.

Figure 12 depicts the empirical cloudiness utility function. This follows a third-degree polynomial curve, attributing a 100% score to 0–25% cloud coverage and gradually falling to zero. This utility function is valid for cloud coverage percentages at a range of 0–100%.

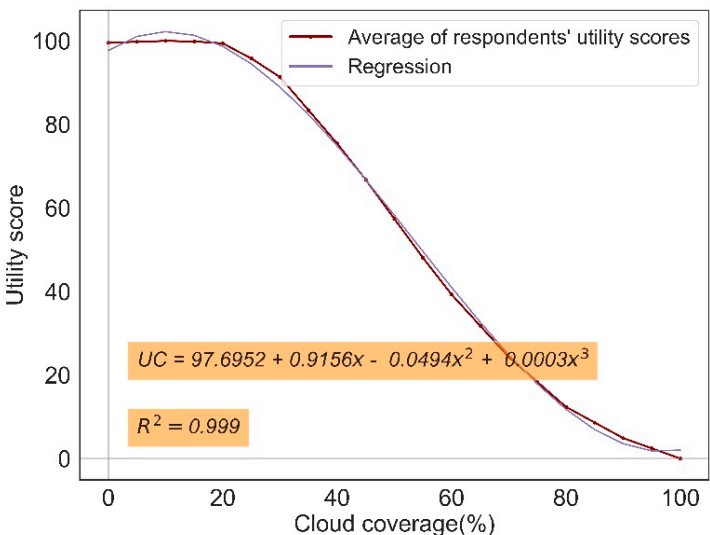

**Figure 12.** Empirical utility function for cloudiness in urban tourism.

The formula of the polynomial for the cloudiness utility function is

$$UC = 97.6952 + 0.9156C_n - 0.0494C_n^2 + 3 \cdot 10^{-4}C_n^3 \tag{29}$$

where $C_n$ is the cloud coverage (in %) for which we wish to calculate the utility score.

### 4.3.5. Relative Humidity

The formula shown in Figure 13 shows the utility function for urban tourists of the relative humidity, as derived from the present empirical study.

As revealed from Figure 13, a relative humidity value between 30% and 40% is assessed as ideal, while the utility score diminishes as we move towards the extremes (0% and 100%). More specifically, the relative humidity levels considered as acceptable by more than 50% of the respondents lie in the range of 20–60%, which agrees with the global bio-meteorological literature ([64] and references within). However, there are limited research on the preferred outdoor relative humidity values for urban tourism. This is also evident from numerous studies that revealed a low significance of relative humidity for outdoor activities [51,53,65]. This apparently low impact of relative humidity on tourist satisfaction could be explained by the availability of adaptation measures that one can apply to deal with such adverse conditions (e.g., avoid a specific location altogether, move indoors, adjust clothing, etc.).

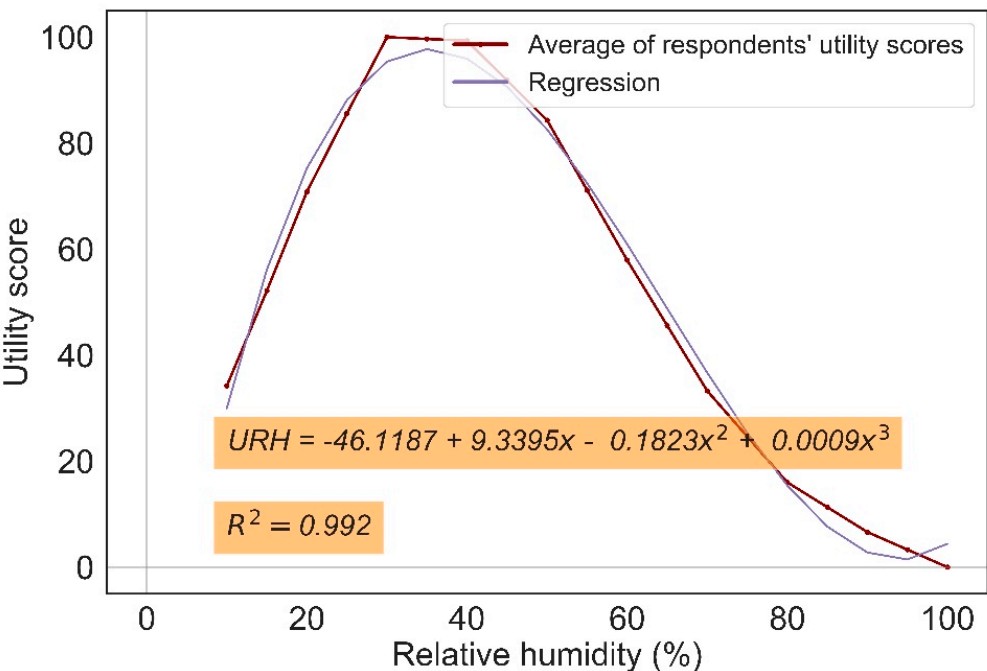

**Figure 13.** Empirical utility function for relative humidity in urban tourism.

The equation of the utility function is a third-degree polynomial with a coefficient of determination of $R^2 = 0.992$ and is valid for relative humidity values of 10–90%

$$URH = -46.1187 + 9.3395RH_n - 0.1823RH_n^2 + 9 \cdot 10^{-4}RH_n^3 \tag{30}$$

where $RH_n$ is the relative humidity (in %) for which we wish to calculate the utility score.

*4.4. Aggregate Utility Score Function—UCCI Index*

Based on the findings presented in the previous sections, we have developed a composite index, the Urban Climate Comfort Index (UCCI), which aggregates the empirical utility functions for the different climate variables (Section 4.3) and the corresponding weight of each variable (Section 4.2). The formula of the UCCI index is

$$UCCI = 0.245UT_n + 0.218UR_n + 0.191UWn + 0.153UC_n + 0.193URH_n \tag{31}$$

Since the UCCI index has been created through a survey carried out in a Mediterranean urban location, namely Athens (Greece), we consider it to be suitable for urban tourism. Furthermore, the UCCI has been developed based on an international population sample visiting a Mediterranean city during summer. Therefore, the index could be applicable to other Mediterranean urban locations or locations with similar climatic characteristics, although such an extended applicability needs to be tested through further empirical studies.

**5. Discussion**

The current empirical study aims to generate a tool that illustrates the suitability of a given destination for urban tourism in terms of its climatic characteristics. The study is based on a tourist survey that took place at the Acropolis Museum of the Greek capital of Athens during the summer of 2019. The participants were asked to rate their climatic preferences during their urban vacations regarding five climatic variables, namely, ambient air temperature, rainfall duration, windiness, cloudiness, and relative humidity. These variables were chosen since they have been consistently found to influence tourist satisfaction.

The respondents were initially asked to rate the importance of each of the five different climatic variables during their summer urban vacations and the weights of the different climatic variables were calculated after normalizing the mean results from all answers. The results showed that ambient air temperature receives the highest importance percentage (24.5%) for urban tourism, followed by rainfall duration (21.8%). The above conclusion agrees with most of the literature, stating that these two variables are the most important (Scott et al. (2008) [50], Georgopoulou et al. (2018) [23]), with rainfall duration being slightly more significant than temperature in coastal environments (e.g., beach tourism) as in the studies of Moreno (2010) [51] and Dubois et al. (2016) [39]. Cloudiness seems to be the least important climatic parameter for urban tourism (15.4%), which may be because it is mainly a weather element of an aesthetic nature.

At a second stage, urban tourists were asked to indicate their ideal value (or range of values) for the above-mentioned climatic variables, as well as the unacceptable values (or range of values). A temperature range of 24–28 °C, a rainfall duration of 1 h or less, a light breeze of 1–9 km/h, a cloud cover of 25% or less, and a relative humidity level of approximately between 30% and 40% were the resulting ideal values. Similarly, unacceptable values consisted of an air temperature below 15 °C and above 35 °C, a rainfall duration of 3 h or more per day, a wind speed of 39 km/h or more, a cloudiness of 90% or more, and a relative humidity of 10% or less and 60% or more. The above-mentioned values agree to a great extent (with very slight discrepancies) with Scott et al. (2008) [50], Rutty and Scott (2010) [57], and Scott et al. (2016) [20], which are all studies concerning tourism in urban destinations. Next, the aggregate theoretical utility functions for each climatic variable were calculated through a linear regression fit method. The UCCI index is formed from weight-summing the respective climatic theoretical utility functions into one coherent aggregate formula.

At this point we should mention that UCCI, although useful for assessing the climate comfort of an urban tourist destination, is inevitably subject to limitations. One possible factor influencing the accuracy of climate indices is the choice of climatic variables included in an index as well as how these variables are incorporated in this index. For example, one could also include atmospheric composition variables (e.g., concentration of air pollutants) in the development of a tourism climate index for urban areas. Moreover, different studies make use of a different representation of some weather elements, such as, for example, using sunshine duration instead of cloud coverage or wet-bulb temperature instead of ambient air temperature. However, in our view, the variables chosen in the present study are straight-forward, easy to comprehend, and their values are usually included in datasets provided by meteorological stations and weather forecasts.

Another potential constraint of our study is that the links between climate comfort and the five climate variables selected were investigated separately. Therefore, any synergies between the different weather elements have not been studied. For instance, the combined effect of very high temperatures and high relative humidity can greatly impact tourist satisfaction and comfort. Moreover, an additional feature that could have been included in the index would be an overriding scheme, which would alter the weights of the different climate variables according to their values, such as that used in other studies (e.g., in [25]). For example, some tourists might evaluate the wind parameter as more important compared to air temperature in the case of extreme wind speeds.

Finally, the present study is of course ultimately built on personal views of a specific random tourist sample (simple random sampling) and, as such, may be subject to biases. For example, some tourists may have difficulties perceiving what an ambient air temperature of 10 °C feels like or even differentiate between 10 °C and 12 °C. To facilitate these considerations, we informed the participants of the values of the climate variables at the time of the survey so that they could form a cognitive perception of their preferences. This difficulty, however, is inherent to many survey studies. On the other hand, the 250 participants to our survey represented diverse nationalities and ages and, hence, a broad range of views.

## 6. Conclusions

Europe hosts, perhaps, the most developed tourism industry worldwide, but only recently has there been an awakening regarding the need for research exploring the impacts of climate change on urban tourism in the region. The current study proposes a new climate index for urban tourism: the Urban Climate Comfort Index (UCCI). The index integrates critical climate variables for urban tourism and is informed by empirical data drawn from a random sampling survey conducted at the Acropolis Museum in Athens, Greece.

The new climate index is easy to use and comprehend, the variables involved are easily retrievable and the finer daily timescale provides a better accuracy since tourists react to the integrated effects of climatic conditions on a daily basis. On the other hand, the UCCI index is subject to the typical biases of simple random sampling (some members of a population may be selected more systematically than others) and uses a limited amount of climate variables, while their effects are not examined synergistically. Future research could focus on expanding the number of respondents to account for effects such as sub-types of urban tourism (e.g., weather preferences may differ for a tourist interested in visiting cultural heritage sites and a tourist interested in leisure activities), or the difference in age and nationality of the respondents, etc. Finally, an overriding scheme could be introduced to account for the effect of extreme weather events, as well as a scheme accounting for the potential effect of different variables working synergistically (i.e., combined effect of temperature and relative humidity).

Travel and tourism directly depend on climatic conditions and, thus, further changes of the latter will affect, at least to some extent, the attractiveness of urban tourist destinations and, consequently, tourist arrivals and income gained through this economic activity. Therefore, an index that reflects in a reliable way the climatic preferences of urban tourists, combined with climate projections at an appropriate spatial and temporal scale, allows an estimation of the potential risks for urban tourist destinations from climate change. This is particularly important for local and national economies that depend largely on tourism such as Greece. By providing a climate comfort index exclusively for urban tourism, our research outcome hopefully contributes to this end.

**Author Contributions:** Conceptualization, E.G.; data curation, D.K.; formal analysis, D.K.; funding acquisition, D.K.; investigation, D.K.; methodology, E.G. and S.M.; project administration, S.M.; software, D.K.; supervision, E.G. and S.M.; validation, S.M.; visualization, D.K.; writing—original draft, D.K.; writing—review and editing, E.G., S.M. and Y.S. All authors have read and agreed to the published version of the manuscript.

**Funding:** The present study was partly funded by the research program 'THESPIA II' (Development of tools and methodologies for the analysis of future climate projections in Europe/the Mediterranean Basin, the estimation of the impacts of climate change and the assessment of adaptation and mitigation strategies) realized by the Institute for Environmental Research and Sustainable Development (Athens, Greece) of the National Observatory of Athens (NOA), Greece.

**Institutional Review Board Statement:** The study was conducted in accordance with the Declaration of Helsinki, and approved by the Institutional Review Board of National Observatory of Athens in Athens, Greece (case number 1128, 21 September 2017).

**Informed Consent Statement:** Informed consent was obtained from all subjects involved in the study.

**Data Availability Statement:** Not applicable.

**Conflicts of Interest:** The authors declare no conflict of interest.

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
