# Peer review of "Weather Preferences for Urban Tourism: An Empirical Study in the Greek Capital of Athens, Greece"

_atmosphere, doi:10.3390/atmos13020282_

Round 1

Reviewer 1 Report

Dear Authors,

Thank you for reviewing this manuscript.

This manuscript covers the topic of climatic issues and individual concerns in tourism by bringing the case of Athens Greece.

I have completed my review of this manuscript and this paper is likely to be almost ready for publication, but the authors should address some minor issues before processing.

1)Introduction is too long, it should be truncated into 4~5 paragraphs and please add references between line 56 and 85 (paragraph 5 and 6).

2)Lit Review section should also be shortened into 5-6 paragraphs. Please delete some sentences that can be discussed in the discussion and conclusion. 

3)Please consider writing one section of conclusion.

4)In the section of 3.2.5, table1 and the sentences show different wind speeds. Please check the numbers and speed measuring unit.

5)Please re-write the abstract to explain the method clearly.

6)Replace words 'climate and climate change' with 'climatic factors and climate change.'

I feel this manuscript is well-prepared for the publication, and recommend getting accepted after this minor revision.

Author Response

Dear Sir/Madam,

Thank you for your useful insight on our paper entitled "Weather preferences for urban tourism: an empirical study in the Greek capital of Athens, Greece". In the Word file attached you will find a point-by-point response to your comments. We tried to address all issues as best as possible whilst maintaining the rest of the paper in the initial form. We feel your comments have had a positive impact on the final design and content of the paper.

Again thank you.

Best regards,

Kapetanakis Dimitrios

Reviewer 2 Report

The paper entitled "Weather preferences for urban tourism: an empirical study in the Greek capital of Athens, Greece" focuses on the development of an “Urban Climate Comfort Index” (UCCI) which integrates critical climate variables for urban tourism and is informed by empirical data from an urban tourist destination in southern Europe. The paper is well-structured and well-written, it looks robust and logical, and it is executed in good English. In fact, it is quite funny to review a paper on this topic in the times when Athens are covered in snow and experiencing the worst snowfalls in the last 35 years. I appreciate how the authors managed to convey the material and present their findings in a clear and concise way, but I have just a couple of comments and suggestions that might help to further improve the paper:

  1. At the end of the Introduction a clear section-by-section structure of the paper can be explained.
  2. The Literature review in Section 2 might use some form of a table to better summarize and compare the findings.
  3. The survey data collection in Section 3 should be better described: what methods has been used (I suspect it was snowball sampling or even the convenience sampling but this should be explicitly explained). The data selection also constitutes the limitations of research that needs to be addressed and acknowledged. The same issue applies to a very low number of questionnaires (250) that also needs to be discussed.
  4. Figures 1-13 might not need the violet background and gridlines, they would look better without them.
  5. In the Conclusions, the pathways for further research need to be added and highlighted. 

Author Response

Dear Sir/Madam,

Thank you for your useful insight on our paper entitled "Weather preferences for urban tourism: an empirical study in the Greek capital of Athens, Greece". Indeed, the weather these past weeks has been really cold and snowy which is very unusual for Greek weather. I guess the UCCI index would not score high for urban tourists.

In the Word file attached you will find a point-by-point response to your comments. We tried to address all issues as best as possible whilst maintaining the rest of the paper in the initial form. We feel your comments have had a positive impact on the final design and content of the paper.

Again thank you.

Best regards,

Kapetanakis Dimitrios
